# ONYA—The Wellbeing Game: How to Use Gamification to Promote Wellbeing

**Daniel Tolks [1,2,\*], Michael Sailer [3] , Kevin Dadaczynski [1,4], Claudia Lampert [5], Julia Huberty [1], Peter Paulus [1] and David Horstmann [1]**

1   Centre for Applied Health Science, Leuphana University Lueneburg, 21335 Lueneburg, Germany; julia.huberty@stud.leuphana.de (J.H.); paulus@uni.leuphana.de (P.P.); david.horstmann@leuphana.de (D.H.)

2   Institute for Medical Education of the University Hospital, Ludwig Maximilian University of Munich, 80539 Munich, Germany

3   Chair of Education and Educational Psychology, Ludwig Maximilian University of Munich, 80539 Munich, Germany; michael.sailer@psy.lmu.de

4   Department of Nursing and Health Sciences, Fulda University of Applied Sciences, 36037 Fulda, Germany; kevin.dadaczynski@pg.hs-fulda.de

5   Leibniz Institute for Media Research, Hans-Bredow-Institut (HBI), 20148 Hamburg, Germany; c.lampert@hans-bredow-institut.de

\*   Correspondence: daniel.tolks@leuphana.de or daniel.tolks@med.uni-muenchen.de; Tel.: +49-4131-677-7966

**Abstract:** The Wellbeing Game uses game design elements to promote wellbeing. Players document their daily activities in the game and categorize them to one or more of five wellbeing-related factors. The users join teams and can create team events to work together and improve their wellbeing status. The present study aims to review the application and the theoretical base of 'The Wellbeing Game', to adapt it to the German context, and to evaluate its health effects in different settings. Additional aims are to analyze the current state of research regarding the links between health, wellbeing, and gamification and to identify crucial game design elements that have to be implemented in the application in order to address the needs of competence, autonomy, and social relatedness according to the self-determination theory.

**Keywords:** gamification; wellbeing; wellbeing game; self-determination theory; mental health; health; e-health

## 1. Introduction

### 1.1. Five Ways to Wellbeing

In research, two different perspectives on wellbeing are present: hedonic wellbeing and eudaimonic wellbeing. The first focuses on subjective wellbeing and emphasizes life-satisfaction, positive mood, and absence of negative mood [1]. The latter focuses on a broad theoretical basis and considers six elements of wellbeing concerned with self-actualization, which are autonomy, environmental mastery, personal growth, relations with others, the purpose of life, and self-acceptance [2]. The two perspectives are not mutually exclusive. A multidimensional approach, considering a combination of both perspectives, is more adequate [3].

The Five Ways to Wellbeing is a communication framework developed to promote wellbeing in the general population [4]. The five ways (connect, be active, take notice, keep learning, and give) are designed to target the individual directly, to be implemented in daily life, as well as to create space for choice and variety. The framework including the described activities is based on scientific evidence.

Derived from a systematic review, the five ways aim to promote subjective as well as psychological wellbeing [5]:

- "Connect" emphasizes the importance of social relationships. Connecting with others supports social life and protects against mental illness [6].
- "Be active" emphasizes the importance of physical activity. Regular exercise is linked to greater wellbeing and also increases self-efficacy and sense of mastery [7].
- "Take notice" emphasizes the importance of self-awareness and mindfulness. Being aware of personal emotions and sensations as well as reflecting on experiences are related to wellbeing [8].
- "Keep learning" emphasizes the importance of learning and achieving. Learning enhances self-esteem, self-efficacy, and more activity in general [9].
- "Give" emphasizes the role of doing something nice for others. Selfless and meaningful behaviors, such as caring for others, promote wellbeing [10].

The concept of the 'Five Ways to Wellbeing' is widely used to promote wellbeing [11], and its factorial structure could be empirically confirmed [12].

To address the 'Five Ways to Wellbeing' to the general population, more innovative intervention strategies and methods are required. On the basis of previous studies, a gamified approach could be one way to address the five ways effectively.

### 1.2. Gamification for Health and Wellbeing

### 1.2.1. Gamification

Gamification emerged as a trend within the business and marketing sector in the second half of 2010 [13], and its importance in academia is reflected by the growing number of published papers in the field [14]. The term was first used in a blog post by Brett Terrill in 2008 and has since been mentioned in different contexts such as education, commerce, computer science, social networks, and marketing [13].

Gamification is defined as: "[ . . . ] the use of game design elements in non-game contexts" [15]. The underlying concept is to use specific mechanisms and motivational elements of entertainment games in non-gaming contexts in order to improve user behavior and engagement [16,17]. Its growing relevance was facilitated by the progress of digital technology as well as the following accompanying factors: ease of access and affordability, new forms of data tracking, and the prevalence of games as a medium, which is deeply rooted in human culture [18,19].

Werbach and Hunter identified 15 important gamification components including avatars, badges, leaderboards, points, performance graphs, meaningful stories, and teams. The authors also highlighted the interplay of points, badges, and leaderboards called the PBL triad [20].

The psychological effects and mechanisms of the gamification components are closely related to the self-determination theory proposed by Ryan and Deci [21]. The effects of gamification are based on the need for competence, autonomy, and social relatedness [20]. Ryan et al. identified elements that affect the motivational pull of video games, specifically predicting that game features conducive to increased perceptions of autonomy, competence, and relatedness enhance motivation to play. Moreover, the fulfillment of these psychological needs is expected, in turn, to be associated with feelings of presence and changes in wellbeing [22].

Studies show that gamification has a positive effect on motivational and performance aspects [16,17]. This promises a dual improvement in making the activities more pleasant and in ensuring people's long-term engagement and motivation [23].

Research in gamification has evolved from definitions of frameworks and taxonomies to technical papers as well as effect and case studies, towards an actual institutionalizing of a cross-disciplinary field including professorships, educational programs, collected volumes, and academic conferences [24]. According to a survey by Seaborn and Fels, the top fields for applying gamification are education

(26%), health and wellness (13%), online communities and social networks (13%), crowdsourcing (13%), and sustainability (10%) [13]. A large majority (87%) of applied gamification research did not mention or address any theoretical foundation [13]. Furthermore, studies often lack a clear distinction between the concepts of gamification and serious games or other digital learning technologies [25]. Because of its interdisciplinary background, the knowledge is spread between different research communities [26]. Furthermore, the absence of long-term and sustained effects of gamified applications as well as small sample sizes are criticized in the literature [26].

### 1.2.2. Gamification for Health

The use of gamification for health-related purposes is very common and is seen as a promising new approach to health behavior change [27–30]. Compared to other digital health interventions, the gamification approach has seven advantages: (1) increase in intrinsic motivation; (2) broad accessibility through mobile technology and wearables; (3) broad appeal and acceptance of games in general across audiences; (4) broad applicability across health and wellbeing risk factors; (5) cost–benefit efficiency in comparison to the development costs of health games; (6) easy integration into everyday life through daily routines; (7) direct contribution to wellbeing by providing positive experiences [16].

Despite the numerous gamified approaches in the health sector, the number of studies in this field is still limited, although it is now progressing at a fast pace [14,26]. Interventions that use gamification are mostly related to chronic disease rehabilitation (cancer, Alzheimer´s disease, stroke, and obesity), physical activity [16,31], mental health [32], diabetes self-treatment [33,34] and pain relief [35]. Interventions using gamification mostly address self-regulatory behavior change techniques such as feedback and monitoring [36]. The most widely applied game design elements are rewards, leaderboards, and avatars. The majority of cases reported positive effects (59%) regarding affect, behavior, cognition, and user experience. However, the lowest impact was found on cognition. No purely negative effects have been reported [16].

### 1.2.3. Gamification for Mental Health and Wellbeing

Although mental health problems are a major concern for the health care systems, only a few studies focus on the effects of gamification elements on mental health [16,26,32,37]. Gamification has been shown to have positive effects on wellbeing, personal growth, stress, and anxiety [37–39]. Few studies showed mixed results concerning the acceptance of users toward gamification for wellness, relaxation, and concentration interventions [40].

In a systematic review, Johnson et al. showed that gamification can foster affect, behavior, cognition, user experience, and has a positive influence on health and wellbeing [16]. According to the review, the strongest evidence was found using gamification to target behavioral outcomes, such as physical activity and dietary habits. In addition to health behavior, gamification can directly enhance wellbeing by generating positive experiences of basic psychological need satisfaction and other aspects of wellbeing such as positive emotions, engagement, relationships, meaning, and accomplishment [16,41]. In a study by Ryan et al., the authors predict that game features that conduce to increased perceptions of autonomy, competence, and relatedness enhance the motivation to play. Furthermore, the experience of satisfying these psychological needs is expected to be associated with feelings of presence while playing and with short-term changes in wellbeing [22].

An evaluation study of a gamified web application showed a significant positive change in wellbeing of university students and employees of a financial organization in New Zealand compared to a control group [42]. Further, a study by Hull from 2009 [39] showed promising results using therapy video games to improve the wellbeing status of children. The theoretical foundation of wellbeing and gamification shows some interesting similarities. This promising connection should be investigated more closely in further research.

### *1.3. The Wellbeing Game*

To raise awareness and promote wellbeing, the Mental Health Foundation of New Zealand (MHF) developed a web-based application named 'The Wellbeing Game' (TWBG) [43]. In the free of charge application, players document their daily activities and categorize them according to one or more of the five ways to wellbeing [4]. Thereby, players develop awareness of their wellbeing-related activities and are being motivated to engage in such activities more frequently. The users join teams and can create team events to work together and improve their wellbeing status. The time spent on wellbeing-enhancing activities (based on the five ways) is rewarded with points, which can be achieved on both an individual and a team basis. TWBG uses game design elements to motivate the users to participate in the game, such as badges, leaderboards, and point systems. After playing the game, users showed higher awareness for possibilities to promote their wellbeing [44], as well as lower stress levels and better wellbeing, compared to non-players [42]. However, as TWBG was only applied in New Zealand, its transferability to other countries and cultures remains unknown.

## 2. The Present Study

The aim of the present study is to review the status quo of TWBG and its theoretical base, to adapt 'The Wellbeing Game' to the German context, and to evaluate its health effects in different settings. This paper also aims to build a theoretical framework for the application of gamification elements to foster the wellbeing status.

Firstly, the theoretical background of the 'Five Ways to Wellbeing' was examined by a literature review of wellbeing interventions. Secondly, the current application was evaluated regarding user experience and game design elements by an expert group as well as by user focus groups. The status of research in gamification was examined and analyzed for its applicability of 'The Wellbeing Game'.

The results confirm the evidence base of the 'Five Ways to Wellbeing' and thus the theoretical base of 'The Wellbeing Game'. The analysis of the user experience revealed several areas of improvement that target the motivational needs of the users. First, rewarding mechanisms have to be enhanced. Giving users more frequent, visual, and immediate performance feedback by adding performance graphs and linking overall scores to level-ups helps to promote game participation [17,20]. Second, wellbeing-related activities should be promoted more actively by proposing possible activities to engage in and through quests as an additional gamification element [20,45]. Third, interaction elements for teams should be increased. To increase the sense of companionship, commentary and liking functions, as well as team-based quests and the suggestion of team activities, are feasible gamification elements [20,45,46].

The variety of the gamification design elements targets all aspects of the self-determination theory (see Table 1). The experience of personal competence is strengthened by awarding points, badges, and leaderboards, and the experience of personal autonomy by the free choice of avatars, wellbeing activities, and teams. The need for social relatedness is covered by the team membership, team events, leaderboards, and peer interaction elements featuring comments and likes.

**Table 1.** Gamification elements in 'The Wellbeing Game' (TWBG) according to the Self-Determination Theory [21].

| Psychological Need | TWBG Gamification Elements |
|---|---|
| Competence | Points<br>Performance graphs<br>Badges<br>Leaderboards |
| Relatedness | Teams<br>Team events<br>Leaderboards<br>Social network functions |
| Autonomy | Customizable profile design<br>Free choice of activity |

The three aspects of the self-determination theory independently predict both enjoyment and future game-playing behavior and can be assumed to have an impact on the wellbeing status of the user of the TWBG [22]. Achievement and social aspects are emphasized in 'The Wellbeing Game', which are positively associated with all three kinds of need satisfaction. According to the study of Xi and Hamari (2019), achievements are the strongest predictor of both autonomy and competence need satisfaction [47].

## 3. Discussion

The game adaptation will consider cultural and legal specifications, such as data regulation, target group requirements, and media literacy. The fact that mental health is considered a taboo topic in Germany, especially in the working environment, may be an obstacle to reaching target audiences [48]. By highlighting the positive aspects of wellbeing, this aspect is aimed to be reduced.

Specific aspects of the German population will be addressed via user stories and quests related to the living environments and daily routines of the German population.

In comparison to the attitudes of the inhabitants of New Zealand, long-term goal orientation and sustainability is more rooted in the German culture [49]. Therefore, the focus of the game on wellbeing for health could be more feasible for the German population. One obstacle regarding the use of gamified applications for health in Germany is the lower utilization of prevention and health promotion programs [50], which is higher in the New Zealand society. The lower use of media in Germany (2017: 292 min/day) compared to New Zealand (2017: 359 min/day) should also be taken into account as it may affect the acceptance of the game [51].

## 4. Outlook

On the basis of the previously presented findings, the German Wellbeing Game named 'ONYA' will be developed in cooperation with the MHF and will then be applied in different settings (teachers, students, job seekers, University faculty members) in Germany. It will be evaluated in a randomized controlled trial regarding the effects on mental health outcomes.

The Wellbeing Game named 'ONYA' will use the effects of gamification to foster the wellbeing status of the German population. In this project, the authors will also analyze the effects of particular game elements on motivational and performance aspects. The results could provide useful insights into the effects of gamification on wellbeing and may also clarify the theoretical link between gamification and wellbeing.

**Author Contributions:** Conceptualization, K.D.; Methodology, D.H. and D.T.; Formal Analysis, D.H. and D.T.; Resources, P.P., J.H.; Writing-Original Draft Preparation, D.T. and D.H.; Writing-Review & Editing, K.D., M.S., C.L., P.P., J.H.; Project Administration, D.H. and D.T.; Funding Acquisition, K.D., P.P.

**Funding:** The project is funded by the Federal Centre for Health Education within the German Federal Ministry of Health.

**Conflicts of Interest:** The authors declare no conflict of interest.

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
