# Peer review of "ONYA—The Wellbeing Game: How to Use Gamification to Promote Wellbeing"

_information, doi:10.3390/info10020058_

Round 1

Reviewer 1 Report

I recommend this paper as a short overview of the connections between gamification and well being with the promise of a more detailed analysis in a next iteration of the game (within German context). Although I understand the general connection to SDT (and I find it appropriate), I do not think this is fully researched within the paper. I would like to see a deeper connection between SDT and the exact design elements of the game in use with a longer discussion of those connections. As it stands, it seems quite general. However, as I stated, presented as foundation for a plan for further research and implementation, this seems a promising direction for development, and I would look forward to seeing the more detailed data about SDT with users and design more in focus.  

Author Response

Thank you very much for your feedback. We share your opinion about the limitation at the current state of the project. The paper is a extend version of a short paper presented at the SEGAH 2019 conference and shall be published in a special issue of the conference. This is the reason, why the project is described in a more general way and more from a theoretical framework. As you have stated correctly the more details analysis will come in the next iteration of the game.

We tried to intensify the discussion regarding the connection between the self determination theory and gamification elements to improve the discussion.

Reviewer 2 Report

In my opinion this paper cannot be published at this stage. It is very interesting from a theoretical point of view. The analysis about the implication of Gamification in wellbeing is very detailed, but there are no detail about the game design that the authors aim to use in their solution. I suggest to transform the paper in a review of the state of the art of Gamification in wellbeing. It is a good starting point for it, but more details and games should be analyzed. Indeed, also the length of the paper is not adequate to a paper journal. 

Just as a suggestion, I would consider to submit this work (at this stage) to a conference and then improve this work making a systematic review.

Author Response

Respond to the reviewer 2:

Thank you for you very precise feedback. We strongly agree to your comments. In fact the paper is part of a special issues related to the SEGAH conference 2018. We submitted and presented it at the conference. The editors invited us to submit a long paper based on the short paper we wrote for the proceedings of the conference. This is the reason that we submitted the paper in this early state of the project with no results focusing on the theoretical framework. We like your idea to shift the focus towards a review of the art of Gamification in wellbeing, but that would not fit into the special issue.

We tried to describe the game design in more details and also the general idea behind the paper. We hope, that is clearer now for the reader, what the purpose of this paper is.

Reviewer 3 Report

The current manuscript presents a review of the Onya application and its theoretical framework. Another aim of the study is described as evaluation of health effects in different settings. One of my major points refers to that aime stated in the abstract: I was not able to find results of health effects in this paper, if this manuscript is some kind of protocol paper methods have to be described in more detail; as well as operationalisation of the constructs. But, nevertheless it is important to be specific about the aim of the current study. I was wondering about the short discussion, it would be beneficial to enlarge perspectives and implications of this study. I would suggest a table for the review with all details about health related marker, if you review as implicated in the aims. Furthermore, a deeper discussion about the self-determination theory and the gamification elements would improve the discussion section.

Author Response

Respond to the reviewer 3

Thank you for you very precise feedback. We strongly agree to your comments. In fact the paper is part of a special issues related to the SEGAH conference 2018. We submitted and presented it at the conference. The editors invited us to submit a long paper based on the short paper we wrote for the proceedings of the conference. This is the reason that we submitted the paper in this early state of the project focusing on the theoretical framework and is more a protocol paper, as you have correctly stated.

We have modified the research question, so that the purpose of the paper will be clearer. We also described in more detail the methods and how they fit to the research question. We hope, that the aim of the paper is clearly stated. We also intensified the discussion regarding the connection of the self determination theory and the gamification elements to improve the discussion.

Round 2

Reviewer 2 Report

The improvements clarify the aim of the paper. 

Reviewer 3 Report

Authors explained mentioned points adequately.